# A Study on the Development of Real-Time Chamber Contamination Diagnosis Sensors

**DOI:** 10.3390/s25010020

**Published:** 2024-12-24

**Authors:** Junyeob Lee, Kyongnam Kim

**Affiliations:** Department of Energy & Advanced Materials Engineering, Daejeon University, Daejeon 34520, Republic of Korea; dwd0411@gmail.com

**Keywords:** contamination, monitoring, plasma

## Abstract

Plasma processes are critical for achieving precise device fabrication in semiconductor manufacturing. However, polymer accumulation during processes like plasma etching can cause chamber contamination, adversely affecting plasma characteristics and process stability. This study focused on developing a real-time sensor system for diagnosing chamber contamination by quantitatively monitoring polymer accumulation. A quartz crystal sensor integrated with flexible printed circuit boards was designed to measure the frequency shifts corresponding to polymer thickness changes. An impedance probe was also employed to monitor variations in the plasma discharge characteristics. The sensor demonstrated high reliability with a measurement scatter of 2.5% despite repeated plasma exposure. The experimental results revealed that polymer accumulation significantly influenced the plasma impedance, and this correlation was validated through real-time monitoring and scanning electron microscopy (SEM). The study further showed that the sensor could detect the transition point of the plasma state changes under varying process gas conditions, enabling the early detection of potential process anomalies. These findings suggest that the developed sensor system can be crucial for diagnosing plasma and chamber conditions, providing valuable data for optimizing preventive maintenance schedules. This advancement offers a pathway for improving process reliability and extending the operational lifetime of semiconductor manufacturing equipment.

## 1. Introduction

Plasma-based technologies play a crucial role in modern semiconductor device manufacturing. The etching process has become increasingly critical for device miniaturization and advanced integration [1]. A representative etching gas, C_4_F_8_ (octafluorocyclobutane), is widely used for etching insulating layers such as silicon oxide (SiO_2_) and silicon nitride (Si_3_N_4_). During processing, C_4_F_8_ forms CF_x_-based polymers, enabling high selective etching, which is crucial for precisely forming via and trench structures [2,3,4]. This allows the anisotropic etching of oxides and nitrides, facilitating the accurate fabrication of fine device structures. However, C_4_F_8_ has certain limitations. Polymers generated during plasma discharge can accumulate on chamber walls or quartz surfaces, posing the potential risk of unexpected process issues [5,6]. Excessive polymer build-up can alter plasma characteristics, leading to process drift, which compromises process consistency and reduces production yield [7,8,9]. Furthermore, abnormal plasma conditions may result in excessive polymer formation, shorten the maintenance cycle of process components, or cause unforeseen complications.

Therefore, it is crucial to monitor contamination during plasma processes in real-time and perform cleaning operations at appropriate intervals. Currently, diagnostic techniques, including Langmuir probes and Optical Emission Spectroscopy (OES), are widely used to monitor process conditions. Langmuir probes offer high accuracy by directly measuring the electrical properties within the plasma discharge region but face limitations due to contamination and physical interference with the process [10,11]. By contrast, OES is a noninvasive diagnostic tool that identifies plasma species by analyzing the light emitted from the medium. However, it has limited resolution and struggles to accurately quantify subtle process changes [12]. To enhance the reliability of plasma processes, it is necessary to address the limitations of the existing diagnostic methods and develop new sensor technologies capable of real-time monitoring. Particularly, it is crucial to quantitatively assess polymer accumulation during the processes and use these data to predict process changes [13,14,15,16].

This study aims to develop a sensor system for the real-time diagnosis of chamber contamination and quantitative analysis of process-state changes. A polymer sensor was designed to measure the amount of polymer accumulated on the chamber walls and near the chuck. Its performance and reliability were validated through repeated plasma process experiments. Additionally, a VI-probe was used to observe changes in the chamber environment and plasma characteristics simultaneously, thereby providing insights into the early detection of potential process anomalies. This study focuses on advancing foundational technologies to improve process reliability and stability while exploring their potential applicability.

## 2. Experimental Conditions

The chamber system used in this study is illustrated in Figure 1. The plasma source installed at the top operates using an Inductively Coupled Plasma (ICP) method with a spiral-type antenna with a diameter of 180 mm. The source power, which was provided by an RF generator with a frequency of 13.56 MHz, was varied in the range of 100 to 200 W for the experiments. The bottom chuck, with a diameter of 4 in, was kept in a floating state with no RF power applied. The chamber pressure was regulated using a vacuum pump to achieve and maintain an optimal pressure for plasma discharge. A dry pump (Dry HC450, Kashiyama, Saku city, Japan) was employed to achieve low-vacuum conditions, and a turbomolecular pump (TMP 2003LM, Shimadzu, Kyoto, Japan) was used under high-vacuum conditions. Process gases, including Ar, O_2_, and C_4_F_8_, which are commonly used in semiconductor manufacturing processes, were introduced into the chamber through mass flow controllers (MFCs). The gas flow rates were controlled as follows: Ar at 65 sccm, O_2_ at 0–2 sccm, and C_4_F_8_ at 30–50 sccm. An impedance probe (ENI VI-Probe, MKS, Yongin-Si, Republic of Korea) was installed between the antenna and matching network to monitor the power transfer from the plasma source to the chamber. This setup enables the measurement of impedance variations in the discharge plasma, providing insights into the power transfer conditions under varying chamber contamination levels.

In typical process environments, the matching network is programmed to adjust the matching position dynamically to minimize the reflected power. However, in this study, the matching network was set to manual mode, keeping the matching position fixed. This approach was used to measure the changes in the impedance and power transfer efficiency over time as polymer deposition occurred.

The sensor was constructed by attaching a quartz crystal sensor between two FPCB (Flexible Printed Circuit Board) films, each measuring 19 mm × 24 mm. The electrodes on the FPCB films were designed to match the size and position of the electrodes on the quartz crystal sensor, enabling electrical signals from the sensor to be collected using an external monitoring device. Utilizing flexible FPCB films enabled the sensor to be easily attached to various surfaces, including chamber walls and the chuck. A 10 mm diameter exposure area was created at the center of the top FPCB film to facilitate the real-time measurement of frequency changes during the deposition or removal of polymers in this region. The operating principle of the polymer sensor used in this study is as follows. The quartz crystal sensor has a natural resonance frequency of 6 MHz. The frequency decreases when polymers accumulate in an exposed area, whereas the frequency increases when the amount of accumulated polymer decreases. The frequency changes corresponding to the amount of polymer accumulated on the sensor surface were monitored in real-time using an external thickness monitor (SQM-160, Inficon, Ragaz, Switzerland). Continuous monitoring of these changes allowed the analysis of the performance and reliability of the sensor. Using this polymer sensor and measurement system, the quantitative variation in the polymers formed inside the chamber during the plasma processes was measured, and the correlation between these changes and the electrical characteristics of the VI-probe was analyzed.

A silicon wafer sample was placed near the sensor to examine the relationship between the thickness of the accumulated polymer and the frequency change in the polymer sensor. Subsequently, the thickness of the polymer on the wafer was measured and compared with the sensor readings. Polymer thickness measurements on the silicon wafer were conducted using an alpha-step profiler (KLA, AS IQ, Milpitas, CA, USA).

## 3. Results and Discussion

Figure 2 shows the experimental results demonstrating the relationship between the frequency change in the sensor and the actual measured polymer thickness. The process conditions included gas flow rates of 65 sccm for Ar, 1 sccm for O_2_, and 50 sccm for C_4_F_8_, with a process pressure maintained at 4.5 mTorr. No power was applied to the chuck, and only the source power of the top plasma system was set to 200 W. As mentioned previously, the polymer sensor operates based on the high-frequency principle, in which the measured frequency decreases proportionally with the amount of polymer accumulated on the sensor surface. The frequency change, Δfrequency, is defined as the difference between the pre-process and post-process frequencies. The relationship between the frequency change measured by the sensor and the actual thickness of the accumulated polymer was analyzed, and the results are illustrated in Figure 2. A silicon wafer was attached to the chamber wall to measure the polymer thickness accurately, with part of the wafer intentionally exposed for easy removal to measure the step height of the accumulated polymer. Because in situ methods for measuring the polymer thickness on chamber walls are limited, ex situ measurements were performed using an alpha-step profiler. As shown in Figure 2, the polymer thickness measured by the alpha-step profiler increases linearly with a decrease in the frequency measured by the sensor, with thicknesses ranging from tens to hundreds of nanometers. To ensure the reliability of the data, potential errors or measurement inconsistencies were evaluated by repeating the process more than 10 times under identical conditions. The spread of data was calculated using the following formula:$$\mathrm{coefficient}\;\mathrm{of}\;\mathrm{variation}:\;\frac{\sigma }{\mu }\times 100\%\;(\sigma :\;\mathrm{standard}\;\mathrm{deviation},\;\mu :\mathrm{mean}\;\mathrm{value}).$$

The spread was approximately 2.5%, indicating highly consistent and stable data despite repeated plasma exposure, as shown in Figure 2.

These results demonstrate that by observing the frequency changes in the sensor, process variables, such as the rate of polymer accumulation, which indicates contamination within the chamber, can be measured in real-time. Additionally, it is anticipated that the amount of polymer formed on the sensor is closely related to the surface area exposed to the plasma and the plasma density near the sensor surface. Therefore, sensors were attached near both the chamber wall and the chuck to comprehensively measure and analyze the polymer accumulation within the chamber from multiple perspectives.

Figure 3a shows the time-dependent frequency change in the sensor at the attached location, and Figure 3b illustrates the trend of the impedance (|*Z*|) measured during the process using the impedance probe installed between the antenna and the matching box. The process conditions involved gas flow rates of 65 sccm for Ar, 2 sccm for O_2_, and 30 sccm for C_4_F_8_, with a process pressure maintained at 4.5 mTorr. Only the top plasma source power of 200 W was applied, and no power was applied to the chuck. The experiments were conducted under identical process conditions to investigate the effect of polymer accumulation by varying the process time up to 3600 s. The data were collected every 600 s to systematically observe trends.

According to Brett A. Cruden, the characteristics of the plasma can change through physical and chemical interactions between the plasma and chamber surfaces. His study showed that, as the level of chamber contamination increased, analyses using a Langmuir probe and an Electrostatic Quadrupole Plasma (EQP) mass spectrometer revealed a decreasing trend in the floating potential, plasma potential, and ion energy. These changes were also observed as a reduction in the impedance magnitude, measured using an impedance monitor [17]. Changes in chamber conditions, particularly polymer accumulation, can negatively affect process reproducibility and uniformity. To address this issue, impedance changes can be monitored to diagnose variations in plasma conditions within the chamber, enabling real-time assessment of contamination levels. Generally, the impedance *Z* of a plasma source during discharge can be expressed by the following relationship [18]:$$\left|Z\right|={\left|E\right|}^{2}/P.$$

Here, ∣*E*∣ represents the magnitude of the voltage between the electrodes, and *P* is the RF power supplied to the electrodes. This equation indicates that when the RF power is constant, the impedance varies in proportion to the square of the voltage ∣E∣ between the electrodes. Therefore, changes in impedance ∣*Z*∣ are closely related to variations in the electrode voltage, which, in turn, reflect changes in plasma conditions (such as density and electron temperature). The voltage distribution and corresponding impedance change along with the plasma state changes. Variations in plasma characteristics can be tracked by monitoring impedance. This demonstrates that impedance can serve as a critical indicator for diagnosing the dynamic state of plasma in real-time. In this study, the experiments were conducted under the assumption that if plasma conditions remain stable and there are no changes in chamber conditions, both sensors should show consistent measurements over time. As shown in Figure 3, during the initial 7200 s of the process, there were no significant changes in the readings from either the sensor or the impedance probe. However, after 7200 s, both sensors exhibited substantial changes in their measurements. Despite maintaining constant process conditions, the sharp change in impedance readings suggests a significant alteration in the plasma discharge state within the chamber. Additionally, a notable decrease in the amount of polymer accumulating within the chamber was observed at the point where the impedance values measured by the impedance probe began to change dramatically. This indicates that the plasma state has deviated from its normal range, and the discharge condition has changed; that is, the accumulated polymer over time has altered the chamber environment, which can be interpreted as a decrease in power transfer efficiency due to increased reflected power. Additionally, the incomplete discharge may have led to changes in the amount of polymer deposition within the chamber.

Figure 4 shows the results of a study conducted to determine whether changes can be detected using a sensor and an impedance probe, similar to previous experimental results, under conditions where the polymer formation rate varied depending on changes in the plasma discharge environment. The process conditions involved flowing 65 sccm of Ar and 50 sccm of C_4_F_8_ through the MFCs and maintaining a process pressure of 130 mTorr to accelerate the polymer accumulation rate compared with previous processes. Power was not applied to the chuck. The thickness variation in the polymer in the chamber was measured in real-time using a polymer thickness sensor, and the impedance was measured and analyzed using an impedance probe at intervals of 1000 s.

Both the sensor readings and the plasma impedance values showed minimal changes up to 6000 s. However, after 6000 s, significant variations in the slopes of both values were observed. This indicates that, similar to the previous experimental results, the plasma state begins to change once the polymer accumulation in the chamber reaches a critical level. As the process progressed, a consistent decrease in the slope was observed. This suggests that the presence of a certain amount of polymer in the chamber can alter the plasma discharge characteristics and this degradation persists under conditions that promote further polymer formation.

Therefore, this study demonstrated that the developed polymer thickness sensor can effectively diagnose the plasma state.

Figure 5a shows the real-time monitoring results of the polymer accumulation inside the chamber under varying process gas ratios. The process conditions involved introducing C_4_F_8_ (0–50 sccm), Ar (10 sccm), and O_2_ (0–65 sccm) through the MFC while maintaining a process pressure of 2.5 mTorr. Power was not applied to the chuck, and only the upper plasma source power was set to 300 W.

Varying the gas flow rates and ratio of the process gases under the same pressure demonstrated that the sensor could detect and measure changes in polymer accumulation through its sensitivity. A silicon wafer was placed next to the sensor inside the chamber to allow for post-process polymer analysis using a scanning electron microscopy (SEM).

CF polymers increased as the flow of C_4_F_8_ increased, owing to an increase in CF_x_ radicals, whereas polymer formation decreased with an increase in O_2_ flow [19]. A significant decrease in the frequency shift in the sensor was observed in region (a), where only the C_4_F_8_ gas was used. However, with the addition of Ar gas, as the C_4_F_8_ ratio decreased and dissociation increased, the frequency shift became less pronounced. Post-process SEM analysis confirmed a substantial accumulation of the polymer on the surface.

In region (b), when Ar and O_2_ were added to C_4_F_8_, the sensor frequency stopped decreasing and began to increase, indicating the removal of the polymer from the sensor surface. When oxygen is introduced into the plasma, its strong reaction with carbon facilitates the rapid removal of CF_x_ polymers [20,21]. During subsequent processes, when C_4_F_8_ was turned off, and only Ar and O_2_ were used for plasma discharge, the sensor frequency increased sharply, indicating rapid polymer removal. When the plasma was discharged using only O_2_, the polymer removal rate further accelerated, and the frequency plateaued, indicating the complete removal of the polymer from the sensor.

The SEM images in Figure 5a confirm that the thick polymer layer was entirely removed, thus restoring the original silicon surface. These results demonstrate that the sensor used in this study responds sensitively to the type and ratio of the process gases, allowing for the diagnosis of polymer changes under various process conditions. Furthermore, the state of complete polymer removal inside the chamber was indicated by the sensor frequency returning to its baseline value. To identify the types of substances present on the surface observed by SEM, chemical composition analysis was performed using X-ray Photoelectron Spectroscopy (XPS). And the results were shown in Figure 5b. XPS is based on the photoelectron effect, where X-rays are directed at the sample, and the kinetic energy of the emitted electrons is measured. This allows for the calculation of the binding energy of the electrons, enabling the analysis of elemental composition and chemical states [22]. In particular, the process using C_4_F_8_ gas leads to the formation of various CF_x_ polymers, which can be confirmed through XPS analysis [23,24]. The analysis of the XPS data measured in this experiment revealed that as the polymer was formed, the CF series peaks significantly increased during the process (a). After the process concluded, the polymer formed on the surface was removed, causing the CF series peaks to nearly disappear, resulting in peak values similar to those of the reference. These results indicate that the polymer in the chamber was removed during the cleaning process, and they also suggest that this cleaning process can be distinguished and accurately measured and analyzed using the sensor employed in this study.

## 4. Conclusions

In this study, sensors were installed on the chamber wall and chuck to quantitatively measure changes in polymer accumulation during the process. These polymer changes could influence the plasma state, and the specific point of electrical change in the plasma could be identified through variations in the sensor frequency and VI-probe measurements. The measurement scatter of the sensor for quantitative polymer changes was 2.5%, demonstrating high reliability with consistent readings, even after repeated exposure to plasma.

This study also proved that changes in the thickness of the polymer accumulated inside the chamber, depending on the contamination level, could be observed through the frequency variation in the sensor. As the process continued, the plasma state transition could be pinpointed by abrupt changes in the impedance measured by the VI-probe and the sensor frequency at specific times.

These findings highlight that the developed sensor can diagnose plasma changes caused by chamber contamination. Furthermore, by accumulating data on the correlation between the plasma state and the sensor frequency variation, this study provides valuable insights for establishing and optimizing the preventive maintenance schedule of process chambers.

## Figures and Tables

**Figure 1 sensors-25-00020-f001:**
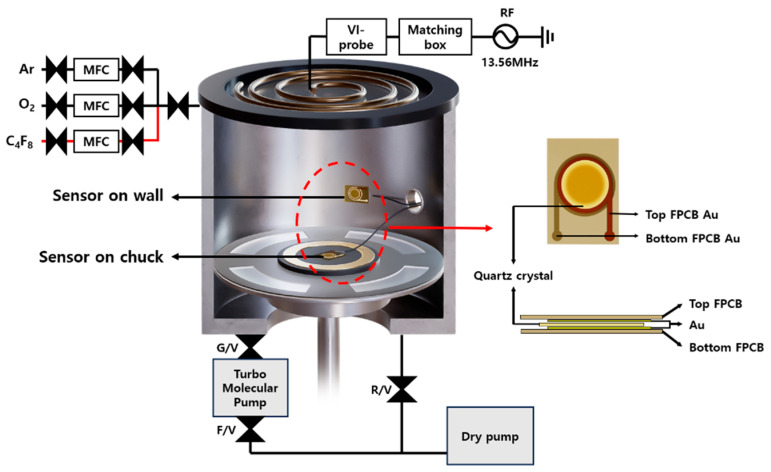
The structure and description of the experimental equipment and the sensor used.

**Figure 2 sensors-25-00020-f002:**
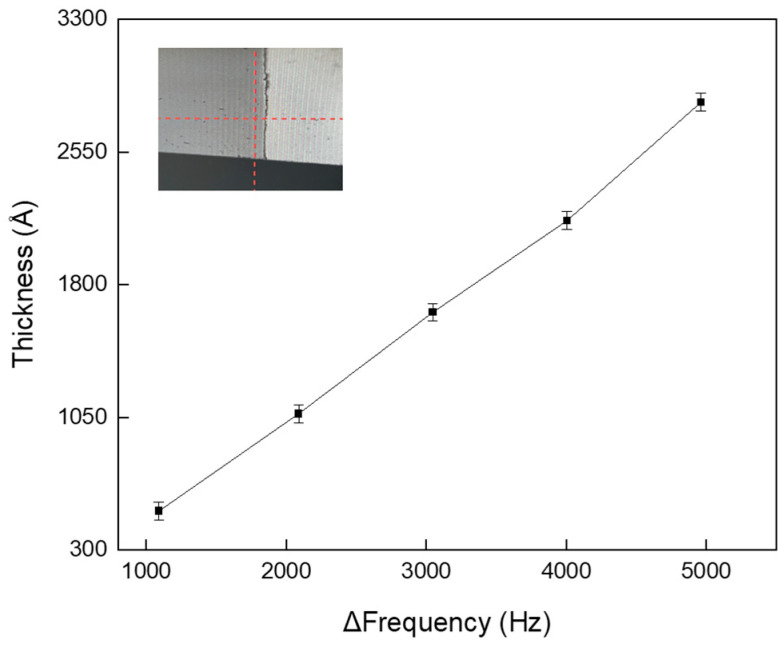
Relationship between sensor frequency change and actual measured polymer thickness.

**Figure 3 sensors-25-00020-f003:**
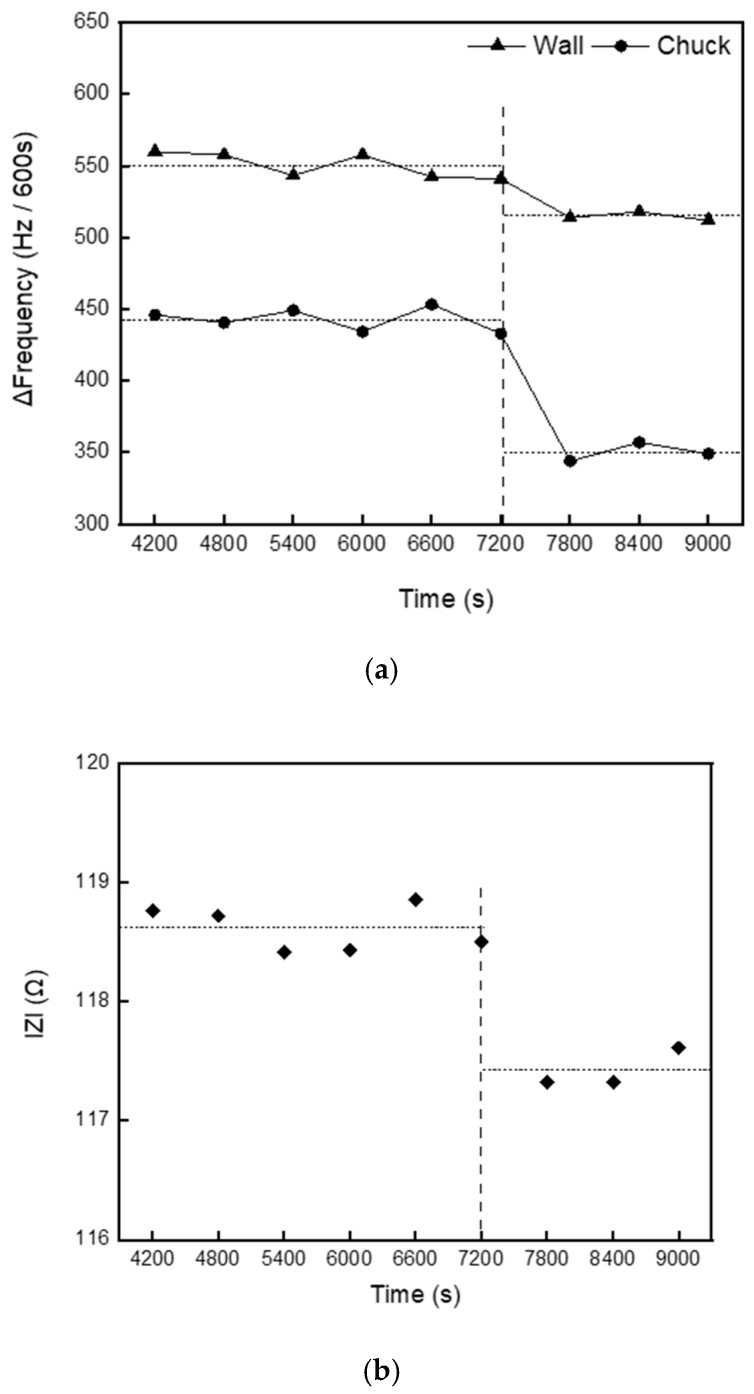
(**a**) Frequency change with time at the sensor location, (**b**) Variation of |*Z*| measured by the impedance probe.

**Figure 4 sensors-25-00020-f004:**
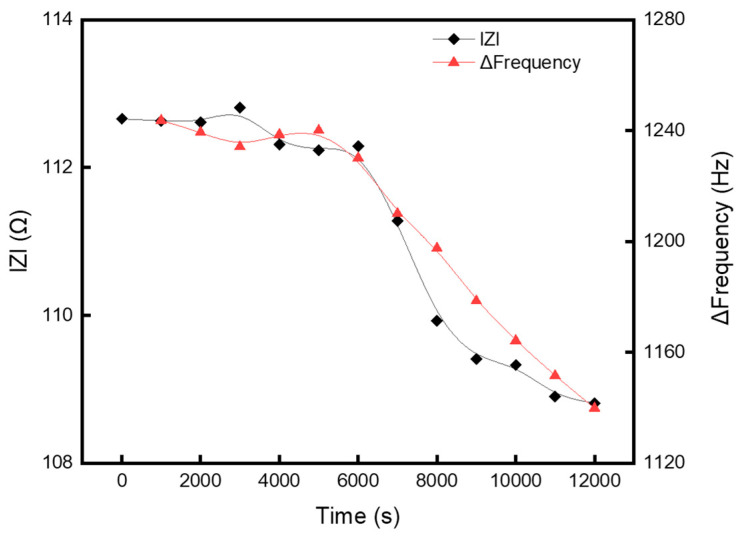
Comparison of frequency shift and impedance variation of the sensor during extended processing.

**Figure 5 sensors-25-00020-f005:**
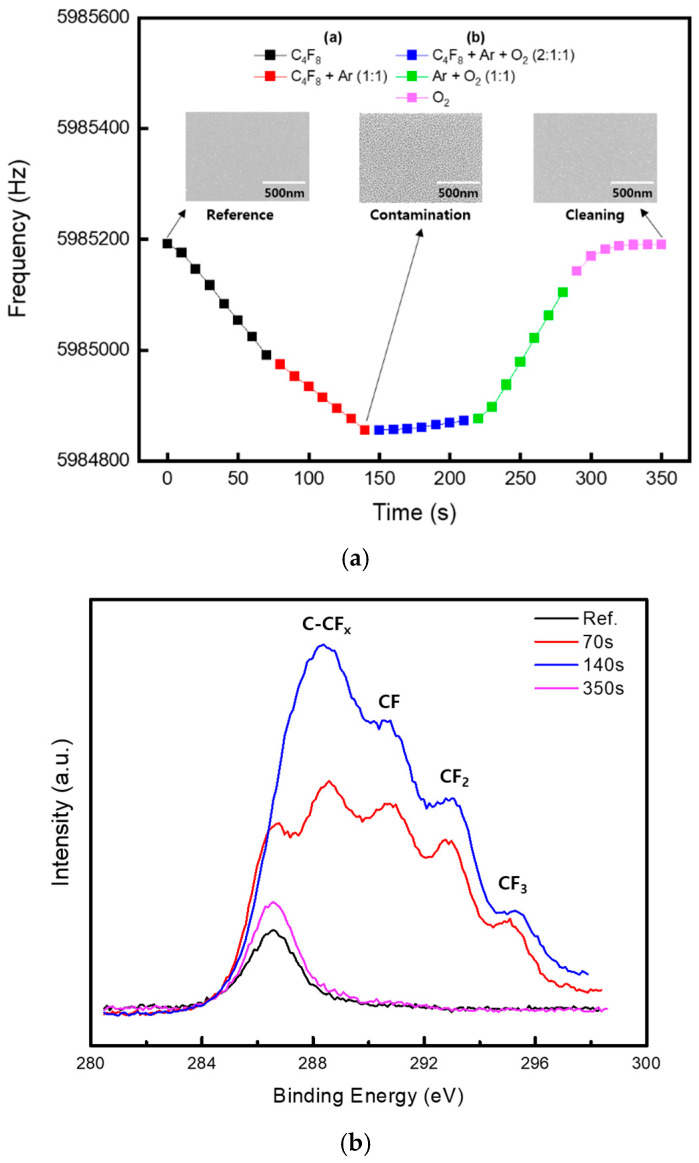
(**a**) Shows the change in frequency according to variations in the plasma environment. And (**b**) presents the results of XPS analysis of the polymer composition deposited on the sensor surface.

## Data Availability

Data are available upon request.

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
