# Peer review of "A Study on the Development of Real-Time Chamber Contamination Diagnosis Sensors"

_sensors, 2024, doi:10.3390/s25010020_

Round 1

Reviewer 1 Report

Comments and Suggestions for Authors

Q1. While Figure 1 is explained in the text, it is necessary to label specific positions in the image. Particularly, the sensor section, electrodes, chuck, etc., should be identified with text boxes. Are there additional setups for the chamber? The figure only shows the ICP-type chamber, but the complete experimental setup, including the dry pump, turbo pump, MFC, and gases, should also be illustrated.

Q2. Figures 1 and 2 do not appear significant enough to warrant separate illustrations. Enlarging the sensor part of Figure 1 and incorporating Figure 2 into it would improve the overall layout.

Q3. Figure 3 includes an error bar, but the graph should explicitly display the n value (n=10). Additionally, the measurement figure shown in the inset should include a scale bar.

Q4. The author explains that polymers were formed and removed, and SEM analysis was performed, but no compositional analysis results are provided. If SEM imaging is done, EDS should be included, and it would be necessary to perform XPS analysis to identify the exact chemical components.

Q5. The SEM image inset in Figure 6 also requires a scale bar.

Q6. Considering the overall length of the manuscript, reducing the number of figures to four seems sufficient. Combining related figures could improve the overall organization of manuscript.

Author Response

First of all, I would like to thank the reviewers for their feedback on this paper. Here are my responses to the review comments. Thank you.

Reviewer 1.

Q1. While Figure 1 is explained in the text, it is necessary to label specific positions in the image. Particularly, the sensor section, electrodes, chuck, etc., should be identified with text boxes. Are there additional setups for the chamber? The figure only shows the ICP-type chamber, but the complete experimental setup, including the dry pump, turbo pump, MFC, and gases, should also be illustrated.

Answer> As the reviewer commented, illustrations and descriptions of the experimental equipment have been added. In addition to the main body of the equipment, explanations of the vacuum system, including the pump and gas injection system, have also been included

Q2. Figures 1 and 2 do not appear significant enough to warrant separate illustrations. Enlarging the sensor part of Figure 1 and incorporating Figure 2 into it would improve the overall layout.

Answer> As the reviewer commented, to enhance clarity and facilitate understanding of the structure and methodology of the experimental equipment, Figures 1 and 2 were combined into a single illustration, and the corresponding content in the manuscript was revised accordingly."

Q3. Figure 3 includes an error bar, but the graph should explicitly display the n value (n=10). Additionally, the measurement figure shown in the inset should include a scale bar.

Answer> As the reviewer commented, the size of the symbols was adjusted to enhance the clarity of the error bars, Additionally, the image inside the inset represents the alpha-step that specifies the thickness of the polymer, and a scale bar could not be included.

Q4. The author explains that polymers were formed and removed, and SEM analysis was performed, but no compositional analysis results are provided. If SEM imaging is done, EDS should be included, and it would be necessary to perform XPS analysis to identify the exact chemical components.

Answer> As the reviewer commented, to identify the substance deposited on the sensor surface as a polymer and to determine its exact chemical composition, XPS analysis was conducted, and the results were included in Figure 5(b). And the corresponding content in the manuscript was revised accordingly.

Q5. The SEM image inset in Figure 6 also requires a scale bar.

Answer> As the reviewer commented, the scale bar has been added to the figure.

Q6. Considering the overall length of the manuscript, reducing the number of figures to four seems sufficient. Combining related figures could improve the overall organization of manuscript.

Answer> As the reviewer commented, I have reduced the number of figures from six to five. While I considered combining Figures 3 and 4, following the reviewer's suggestion to create a total of four figures, I could not merge them because I felt that each figure represents a distinct topic.

Reviewer 2 Report

Comments and Suggestions for Authors

Referee’s comments on ‘A Study on the Development of Real-Time Chamber Contamination Diagnosis Sensors’ by Junyeob Lee and Kyongnam Kim

The authors have used resonant frequency shifts with an oscillating quartz crystal to monitor polymer coating in an inductively coupled plasma and compared the measurements to the deduced plasma impedance.  The frequency changes were calibrated (figure 3) with the use of a silicon wafer where the polymer thickness was measured after exposure externally to the discharge chamber.  The in-situ thickness monitoring provided evidence of changes in plasma conditions (figure 4 and 5) and showed that changes of plasma gases could remove coated polymer (figure 6).    The work contributes to the employment of plasma coating in semiconductor manufacture. 

Some minor suggestions:

1.        The first equation on p4 could be expressed as simply: ‘The standard deviation of the repeated measurements was 2.5% of the mean.’    Accuracy figures also should not be quoted to three significant figures (i.e. don’t use 2.53%).   

2.       The authors should quote a reference for the second equation on p4 giving the impedance expression. 

Author Response

First of all, I would like to thank the reviewers for their feedback on this paper. Here are my responses to the review comments. Thank you.

Reviewer 2.

The authors have used resonant frequency shifts with an oscillating quartz crystal to monitor polymer coating in an inductively coupled plasma and compared the measurements to the deduced plasma impedance.  The frequency changes were calibrated (figure 3) with the use of a silicon wafer where the polymer thickness was measured after exposure externally to the discharge chamber.  The in-situ thickness monitoring provided evidence of changes in plasma conditions (figure 4 and 5) and showed that changes of plasma gases could remove coated polymer (figure 6).    The work contributes to the employment of plasma coating in semiconductor manufacture. 

Some minor suggestions:

  1. The first equation on p4 could be expressed as simply: ‘The standard deviation of the repeated measurements was 2.5% of the mean.’    Accuracy figures also should not be quoted to three significant figures (i.e. don’t use 2.53%).   

Answer> As the reviewer commented, the standard deviation of the repeated measurements was corrected to 2.5% of the mean.

  1. The authors should quote a reference for the second equation on p4 giving the impedance expression. 

Answer> As the reviewer commented, we have placed the reference in their correct location and incorporated the revisions into the text